# Expression of Steroid Receptor RNA Activator 1 (SRA1) in the Adipose Tissue Is Associated with TLRs and IRFs in Diabesity

**DOI:** 10.3390/cells11244007

**Published:** 2022-12-11

**Authors:** Shihab Kochumon, Hossein Arefanian, Sardar Sindhu, Reeby Thomas, Texy Jacob, Amnah Al-Sayyar, Steve Shenouda, Fatema Al-Rashed, Heikki A. Koistinen, Fahd Al-Mulla, Jaakko Tuomilehto, Rasheed Ahmad

**Affiliations:** 1Department of Immunology & Microbiology, Dasman Diabetes Institute, Dasman 15462, Kuwait; 2Animal and Imaging Core Facilities, Dasman Diabetes Institute, Dasman 15462, Kuwait; 3Department of Medicine, University of Helsinki and Helsinki University Hospital, 00029 Helsinki, Finland; 4Department of Public Health and Welfare, Finnish Institute for Health and Welfare, 00271 Helsinki, Finland; 5Minerva Foundation Institute for Medical Research, 00290 Helsinki, Finland; 6Department of Genetics and Bioinformatics, Dasman Diabetes Institute, Dasman 15462, Kuwait; 7Department of Public Health, University of Helsinki, 00100 Helsinki, Finland; 8Saudi Diabetes Research Group, King Abdulaziz University, Jeddah 21589, Saudi Arabia

**Keywords:** steroid receptor RNA activator 1/SRA1, TLRs, IRFs, adipose tissue, obesity, type-2 diabetes, inflammation

## Abstract

Steroid receptor RNA activator gene (SRA1) emerges as a player in pathophysiological responses of adipose tissue (AT) in metabolic disorders such as obesity and type 2 diabetes (T2D). We previously showed association of the AT SRA1 expression with inflammatory cytokines/chemokines involved in metabolic derangement. However, the relationship between altered adipose expression of SRA1 and the innate immune Toll-like receptors (TLRs) as players in nutrient sensing and metabolic inflammation as well as their downstream signaling partners, including interferon regulatory factors (IRFs), remains elusive. Herein, we investigated the association of AT SRA1 expression with TLRs, IRFs, and other TLR-downstream signaling mediators in a cohort of 108 individuals, classified based on their body mass index (BMI) as persons with normal-weight (N = 12), overweight (N = 32), and obesity (N = 64), including 55 with and 53 without T2D. The gene expression of SRA1, TLRs-2,3,4,7,8,9,10 and their downstream signaling mediators including IRFs-3,4,5, myeloid differentiation factor 88 (MyD88), interleukin-1 receptor-associated kinase 1 (IRAK1), and nuclear factor-κB (NF-κB) were determined using qRT-PCR and SRA1 protein expression was determined by immunohistochemistry. AT SRA1 transcripts’ expression was significantly correlated with TLRs-3,4,7, MyD88, NF-κB, and IRF5 expression in individuals with T2D, while it associated with TLR9 and TRAF6 expression in all individuals, with/without T2D. SRA1 expression associated with TLR2, IRAK1, and IRF3 expression only in individuals with obesity, regardless of diabetes status. Furthermore, TLR3/TLR7/IRAK1 and TLR3/TLR9 were identified as independent predictors of AT SRA1 expression in individuals with obesity and T2D, respectively. Overall, our data demonstrate a direct association between the AT SRA1 expression and the TLRs together with their downstream signaling partners and IRFs in individuals with obesity and/or T2D.

## 1. Introduction

Obesity is known as a complex disease due to an excessive amount of body weight, and mainly the body fat associated with expansion and function of white adipose tissue. The expanded white adipose tissue in individuals with obesity produces a wide range of adipocytokines, such as proinflammatory cytokines, chemokines, hormones, and similar mediators [1,2,3,4]. Adipocytes, resident monocytes/macrophages, and other cell populations in the adipose tissue are actively involved in production and secretion of adipocytokines [5,6,7,8]. It is speculated that the mechanisms of obesity-induced insulin resistance are initiated and stimulated by these adipocytokines, mainly by proinflammatory cytokines in the white adipose tissue [9,10,11,12,13].

Toll-like receptors (TLRs) are the surface, innate immune receptors that identify the pathogen-associated molecular patterns (PAMPs) to activate and initiate inflammatory responses. In humans, 11 different TLRs have been so far identified [14,15]. Structurally, TLRs have an extracellular, leucine-rich repeat (LRR) domain which recognizes PAMPs, and a cytoplasmic Toll/IL-1 (TIR) domain that activates TLR-downstream signaling after ligand binding with its cognate TLR. LRR and TIR are involved in the recognition of PAMPs and activation of downstream adaptor proteins and signaling molecules, including myeloid differentiation factor 88 (MyD88), interleukin-1 (IL-1) receptor-associated kinases (IRAKs), and tumor necrosis factor receptor-associated factor (TRAF)-6, respectively [15]. After the activation of these adaptor proteins, stimulation of multiple pathways is initiated such as extracellular signal-regulated kinase (ERK), c-Jun N-terminal kinase (JNK), p38 mitogen-activated protein kinases (MAPK), NF-κB, and interferon regulatory factors (IRFs) pathways. The activation of NF-κB signaling results in the up-regulation of inflammatory markers including cytokines, chemokines, adhesion molecules, type-I interferons (IFN-α/β), tumor necrosis factor (TNF)-α, and IL-6) which offset the homeostasis between beneficial host innate immune responses and immunopathology [16,17,18,19].

TLRs such as TLR2, TLR3, TLR4, TLR7, TLR9, and TLR10 have been identified in multiple immune cell populations within the adipose tissue namely adipocytes, monocytes, and macrophages. Each of these TLRs has distinct ligands such as free fatty acids (FFAs), lipids, lipoproteins, nucleic acids, and proteins which indicate the potential role of innate immune TLRs as nutrient sensors involved in obesity [20,21]. The important roles of TLR signaling cascades in adipose tissue inflammation and impairment in obesity and type-2 diabetes (T2D) have been addressed and reported by us and others [22,23,24]. Expression changes in TLR1, TLR2, TLR4, TLR5, TLR6, TLR7, TLR8, TLR9, and TLR10 have been often reported in obesity, metabolic syndrome, inflammation and insulin resistance, T2D, and related complications such as cardiovascular disease, diabetic nephropathy, and atherosclerosis [23,24,25,26,27,28,29,30,31,32,33,34]. Specifically, TLR4/TLR2 have emerged as metabolic sensors of lipopolysaccharide (LPS) and saturated free fatty acids (sFFAs), both of which are abundantly found in individuals with obesity and T2D [35].

Steroid receptor RNA activator 1 (SRA1) was originally identified as an intergenic long non-coding RNA (lncRNA) which acts as an RNA coactivator of nuclear receptors to enhance steroid receptor-dependent gene expression [36]. It binds with DNA via interactions with other proteins that bind directly or indirectly with the DNA. Therefore, serving as a natural organizer by regulating physiological processes that dictate the epigenetic modifications including changes in chromatin and gene expression [37,38,39]. Increased SRA1 expression in the human liver, skeletal muscle, and in the white/brown adipose tissues as key organs regulating metabolic homeostasis, compared to other tissues, has been documented [36,40,41]. We recently showed that in individuals without T2D, adipose *SRA1* expression was significantly higher in obese people compared with normal weight people and the adipose tissue *SRA1* expression associated directly with metabolic markers including body mass index (BMI), percentage of body fat (PBF), serum insulin, homeostasis model assessment of insulin resistance (HOMA-IR), proinflammatory cytokines and chemokines or their receptors including C-X-C motif ligand-9 (CXCL9), CXCL10, CXCL11, TNFα, transforming growth factor-β (TGFβ), IL2RA, and IL18, but inversely with CCL19 and CCR2 expression. We further showed that TGFβ and IL18 were independent predictors of *SRA1* expression in individuals without T2D, while TNFα and IL2RA were the independent predictors in individuals with T2D. TNFα also predicted *SRA1* adipose expression in both normal weight and obese populations, regardless of diabetes status. Taken together, this study revealed specific association patterns of the adipose *SRA1* expression with diverse immune markers, most of them being inflammatory by nature [42].

In the current study, we examined whether the human adipose tissue SRA1 expression associated with TLRs expression, their adaptor proteins and TLR-downstream signaling molecules including NF-κB, IRF3, 4, and 5 in obesity and/or T2D.

## 2. Materials and Methods

### 2.1. Study Population and Anthropometry

This study included 108 participants who were classified based on BMI as persons with normal weight (NW) (BMI < 25 kg/m^2^, n = 12), overweight (25 ≤ BMI < 30 kg/m^2^, n = 32), or obese (BMI ≥ 30 kg/m^2^, n = 64). Among these participants, 53 were with and 55 were without T2D. Diagnosis of T2D was based on fasting blood glucose and glycated hemoglobin (HbA1c) levels. HbA1c levels of <5.7, 5.7–6.4%, and >6.4% represented normal, prediabetes, and diabetes status, respectively. Height, weight, waist and hip circumferences, BMI, and PBF were measured and calculated as previously described [42]. For the assessment of insulin resistance (HOMA-IR) and insulin sensitivity, fasting blood glucose and insulin levels were used to determine HOMA index [43]. This study was approved by ethics committee of the Dasman Diabetes Institute, Kuwait in line with the ethical guidelines of the Declaration of Helsinki (Grant#: RA 2010-003). The written informed consent was obtained from each participant at the time of enrolment in the study. The individuals with chronic diseases of the heart, liver, kidney, lung, or those with type 1 diabetes, immune dysfunction, hematologic disorders, pregnancy, or malignancy were excluded previously stated [42,44]. Clinical and demographic features of the study cohort are presented in Appendix A.

### 2.2. Collection of Subcutaneous Adipose Tissue Samples

Biopsies of the human adipose tissue, about 0.5 g in size, were collected from the abdominal subcutaneous adipose tissue, next to the umbilicus by using sterile surgical technique as described [42,45]. The sample was further cut into smaller pieces, about 50–100 mg in size, and added to RNAlater (Sigma-Aldrich Chemie GmbH, Taufkirchen, Germany) and stored at −80 °C until use for RNA extraction.

### 2.3. Measurement of Metabolic Markers

Peripheral blood was collected from the individuals fasting overnight and the samples were analyzed for metabolic markers such as plasma glucose, insulin, and lipid profiles including triglycerides (TGL), low-density/high-density lipoproteins (LDL/HDL), and total cholesterol using Siemens Dimension RXL Analyzer (Diamond Diagnostics, Holliston, MA, USA). Glycated hemoglobin (HbA1c) was measured by Variant device (BioRad, Hercules, CA, USA).

### 2.4. Quantitative, Real-Time, Reverse-Transcription Polymerase Chain Reaction (RT-qPCR)

Fat samples were used for total RNA collection by using RNeasy kit (Qiagen, Valencia, CA, USA), following the protocol as recommended by the manufacturer. RNA template (0.5 µg) was used for cDNA synthesis using High-Capacity cDNA Reverse Transcription Kit (Applied Biosystems, Foster, CA, USA) as earlier stated [45]. Real-time qRT-PCR was carried out as per the protocol [42,45,46]. Briefly, 50 ng of cDNA was amplified using TaqMan Gene Expression Master Mix (Applied Biosystems, CA, USA) and gene-specific 20 X TaqMan gene expression assays including forward and reverse primers (Appendix A), target-specific TaqMan 5′FAM-labeled and 3′NFQ-labeled MGB probe, using 7500 Fast Real-Time PCR System (Applied Biosystems, CA, USA) as described elsewhere [42]. Target gene expression relative to control (NW fat samples) was determined using comparative Ct method [47] and data were normalized to GAPDH gene expression as described [48,49,50,51,52].

### 2.5. Immunohistochemistry (IHC)

Expression of SRA1, TLR4, IRAK1 or NF-kB was measured in the fat tissue by IHC as described elsewhere [42,53]. Briefly, subcutaneous fat tissue (paraffin-embedded) was cut in 4μm thick sections, deparaffinized by xylene, and rehydrated by serial immersions in 100%, 95%, and 75% ethanol in water. Antigen was retrieved by retrieval solution (pH 6.0; Dako, Glostrup, Denmark) with 8 min boiling and 15 min cooling steps. After 3 washes with PBS, internal peroxidase was blocked by 30 min treatment with 3% H_2_O_2_ and non-specific antibody binding was blocked by 1 h treatment, each, with 5% non-fat milk and 1% BSA. The primary antibody treatment was carried out overnight at room temperature, using anti-human SRA1 rabbit polyclonal antibody (1:800 dilution) (Thermo-Scientific PA5-62145, pH 6.0), rabbit monoclonal anti-IRAK antibody (ab302554, Abcam*^®^* Cambridge, UK), 1:400 dilution of rabbit monoclonal anti-IRF5 antibody (ab181553, Abcam*^®^* Cambridge, UK), 1:1000 dilution of rabbit polyclonal anti-NFkB antibody (ab16502, Abcam*^®^* Cambridge, UK), and 1:1000 dilution of mouse monoclonal anti-TLR4 antibody (ab13556, Abcam*^®^* Cambridge, UK). After washing 3 times using PBS (0.5% Tween), samples were incubated for 1 h with secondary antibody (HRP-conjugated goat anti-rabbit; EnVision™ Kit from Dako, Glostrup, Denmark) and the substrate (3,3′-DAB chromogen) was added to develop color. After washing 3 times in running water, samples were counterstained (Harris hematoxylin), dehydrated by immersion in 75%, 95%, and 100% ethanol in water, clarified by xylene, and mounted in DPX. Later, digital photomicrographs were taken (40X magnification) and regional heterogeneity was assessed in 4 different regions of tissue sample (PannoramicScan, 3DHistech, Budapest, Hungry). The data were expressed as the staining intensity measured in arbitrary units (AU) and analyzed using imageJ software (NIH, Bethesda, MD, USA).

In addition, we also performed H&E staining on lean and obese adipose tissue samples, 5 each, to represent sample quality for standard tissue staining (Appendix A).

### 2.6. Statistical Analysis

The data obtained were presented as mean ± SEM values and analyzed using GraphPad Prism (GraphPad, San Diego, CA, USA) and SPSS (IBM SPSS Inc., Chicago, IL, USA) software. Means between two groups were compared using unpaired *t*-test, and between more than two groups were compared using One-way ANOVA, Kruskal-Wallis and Mann-Whitney tests. Spearman correlation and multivariate regression analyses were performed to determine associations between variables. All *p*-values ≤ 0.05 were considered significant. For multivariate linear regression, the Enter method was used, selecting the variables that significantly correlated with *SRA1* expression as predictor variables and were entered simultaneously to generate the model. *F*-test was used to test whether the independent variables collectively predicted the dependent variable. *R*-squared evaluated how much variance in the dependent variable was accounted for by the set of independent variables. The *p*-value assessed the significance and the β-value identified the magnitude of prediction for each independent variable.

## 3. Results

### 3.1. SRA1 Adipose Expression and Its Association with TLRs, Downstream Signaling Mediators and IRFs Expression in the Study Population

We sought to determine SRA1 protein expression in adipose tissue samples from NW individuals and those with obesity, 10 individuals each, and the data show that the expression was significantly higher in people with obesity compared to their NW counterparts, whether with or without T2D (*p* < 0.0001) (Figure 1).

We next sought to determine the adipose *SRA1* gene expression and assessed its relationship with typical inflammatory sensing and signaling components such as TLRs, their downstream signaling mediators, and IRFs. The expression of these markers was first compared among populations defined as those with NW, overweight, and obesity. To this end, adipose SRA1 mRNA expression differed significantly only between non-diabetic NW and those with obesity individuals (*p* = 0.015) whereas it differed non-significantly between all other BMI groups, with or without T2D (Table 1). Regarding expression of TLRs, their downstream signaling mediators and IRFs, adipose expression of TLR2 (*p* = 0.018), TLR3 (*p* = 0.010), TLR8 (*p* = 0.048), IRAK1 (*p* = 0.047), and IRF5 (*p* = 0.016) was significantly higher in participants with obesity compared to NW participants. IRF5 expression in both overweight (*p* = 0.031) and those with obesity (*p* = 0.016) differed significantly from that of NW participants. Expression of IRF3 (*p* = 0.047), IRF4 (*p* = 0.020), and IRF5 (*p* = 0.039) was significantly higher in individuals with T2D compared with those without T2D. Furthermore, adipose *SRA1* expression was associated with TLR2 (r = 0.218, *p* = 0.036), TLR3 (r = 0.218, *p* < 0.0001), TLR4 (r = 0.226, *p* = 0.027), TLR7 (r = 0.196, *p* = 0.045), NF-κB (r = 0.297, *p* = 0.002), and IRAK1 (r = 0.201, *p* = 0.044) expression in the total (N = 108) study population (Table 2; Figure 2A–F).

Furthermore, to confirm the expression of protein levels in the adipose tissue, we performed immunohistochemistry analysis on TLR4, IRAK1 and NF-kB as representatives. Immunohistochemistry analysis showed that TLR4 (Figure 3A,B), IRAK1 (Figure 4A,B) and NF-kB (Figure 5A,B) were significantly upregulated in individuals with obesity. Our protein data show that SRA1 positively correlated with TLR4 protein (r^2^ = 0.623; *p* = 0.0002; Figure 3C). IRAK1 protein (r^2^ = 0.0649; *p* < 0.0001; Figure 4C) and NF-kB protein (r^2^ = 0.379; *p* = 0.0085; Figure 5C).

### 3.2. Association of Adipose SRA1 Expression (mRNA) with TLRs, Their Signaling Mediators, and IRFs in Individuals Classified as Those with NW, Overweight, and Obesity

Regarding the association between adipose *SRA1* gene expression and meta-inflammatory markers studied (Table 3), we found that *SRA1* correlated with TLR2 (r = 0.317, *p* = 0.017), TLR3 (r = 0.531, *p* < 0.0001), TLR4 (r = 0.311, *p* = 0.022), TLR7 (r = 0.305, *p* = 0.015), TLR9 (r = 0.374, *p* = 0.002), MyD88 (r = 0.324, *p* = 0.010), IRAK1 (r = 0.255, *p* = 0.044), NF-κB (r = 0.454, *p* < 0.001), IRF3 (r = 0.290, *p* = 0.030) and IRF5 (r = 0.321, *p* = 0.010) expression in individuals with obesity. *SRA1* associated inversely with TLR9 expression (r = −0.489, *p* = 0.005) only in overweight group. In NW participants, *SRA1* was associated with MyD88 (r = 648, *p* = 0.043), IRF3 (r = 0.0857, *p* = 0.014), and IRF5 (r = 0.929, *p* = 0.003) expression. Heat map shown in Figure 6.

### 3.3. Association of Adipose SRA1 Expression (mRNA) with TLRs, Their Signaling Mediators, and IRFs in Individuals with/without T2D

In the analysis whether diabetic status affected associations between adipose *SRA1* expression and other markers of inflammation, we found that in people with T2D, *SRA1* expression was associated with TLR3 (r = 0.555, *p* < 0.0001), TLR4 (r = 0.302, *p* = 0.044), TLR7 (r = 0.292, *p* = 0.040), TLR9 (r = 0.398, *p* = 0.003), NF-κB (r = 0.381, *p* = 0.005, MyD88 (r = 0.311, *p* = 0.030), TRAF6 (r = 0.286, *p* = 0.038), and IRF5 (r = 0.288, *p* = 0.043) expression (Table 4; Figure 7). On the other hand, in participants without T2D, *SRA1* was inversely associated with TLR9 (r = −0.290, *p* = 0.034) and TRAF6 (r = −0.318, *p* = 0.019) expression (Table 4).

### 3.4. Analysis of the Independent Associations between SRA1 and TLRs, Their Signaling Mediators, and IRFs

In order to determine independent associations, the markers showing significant associations with adipose *SRA1* expression were further assessed by multivariable stepwise linear regression analysis (Table 5). We found that in the total population (N = 108), TLR3 and IRAK1 independently predicted the adipose SRA1 expression. In participants with T2D (N = 53), TLR3 and TLR9 were identified as the independent predictors of adipose *SRA1* expression, but not in participants without T2D (N = 55). Regression analysis stratified by obesity status revealed the independent association of adipose SRA1 expression with MyD88 in NW (N = 12), and with TLR9 in overweight (N = 32) participants. In individuals with obesity (N = 64), TLR3, TLR7 and IRAK1 were detected as the independent predictors of adipose SRA1 expression.

Multiple linear regression analysis is performed to identify TLRs and their signaling molecules associated with SRA1 as predictor variables.

## 4. Discussion

There is a relative lack of data highlighting the role and significance of long noncoding RNAs in metabolic inflammation and insulin resistance. In the present study, we show that the adipose tissue SRA1 protein expression is significantly increased in individuals with obesity, as compared to their normal weight counterparts, regardless of T2D status. While *SRA1* mRNA expression is significantly elevated, only in people with obesity people without T2D as compared to normal-weight counterparts. We further demonstrate that adipose *SRA1* gene expression is associated with the expression of TLRs, their signaling mediators and IRFs including TLR2, TLR3, TLR4, TLR7, TLR9, MyD88, IRAK1, NF-κB, IRF3, and IRF5. There is a general consensus that obesity is associated with low-grade chronic inflammation, marked by the abnormal production of pro- and anti-inflammatory cytokines and adipokines by the expanding white adipose tissue, which influences changes in adipose tissue expression of the innate immune receptors such as TLRs. The growing evidence now supports that TLRs play a role in inducing a systemic acute phase response characterized by chronic inflammation and oxidative stress [1,2,3,4]. Apart from PAMPs, FFAs also act as TLR agonists which consolidates emerging role of TLRs as metabolic sensors and as receptors of immune-metabolic significance [20]. Regarding nutrient sensing mechanism and ensuing inflammatory responses, TLR signaling is initiated in the TIR domain and the TLR-downstream signaling is propagated through the pathways dependent or independent of MyD88 adaptor protein [16,17]. MyD88-dependent signaling is induced after TLR-ligand engagement and TLR dimerization, leading to recruitment of MyD88 and IRAKs to the TIR domain [54,55]. IRAKs are known as the death domain-containing serine/threonine kinases and adapter proteins which play key roles in signaling pathways of the IL-1 family receptors and TLRs. IRAK1 is activated after phosphorylation by IRAK4 and associates with TRAF6. The IRAK1/TRAF6 complex then engages with the TGFβ-activated kinase (TAK)-1 and TAB-1/2 adapter proteins to yield a macromolecular complex [56]. IRAK1, following its phosphorylation, disengages from the signaling complex and TAK1 activates the inhibitor of NF-κB kinase alpha/beta (IKKα/β), which leads to the phosphorylation, ubiquitination, and degradation of IκBα to allow nuclear translocation of the p65 NF-κB complexes [57]. In parallel, TAK1 phosphorylates MAPKs such as MKK4, MKK3, or MKK6 which, in turn, activate the ERK, JNK, p38 MAPK, NF-κB, and IRFs cascades. Activation of these signaling pathways leads to the increased expression of inflammatory markers including cytokines, chemokines, and adhesion molecules [16,17,18,19,58].

TLRs, especially, the TLR4 and TLR2 have emerged as key players in metabolic inflammation by nutrient sensing of LPS as well as sFFAs, both of which are abundantly found in individuals with obesity/T2D [35]. Increased expression of TLR2 and TLR4 has been found in patients with T2D, highlighting their associations with the pathogenesis of diabetes [24,28]. We previously showed that the elevated adipose gene expression of TLR4, TLR2, and MyD88 in people with obesity/T2D was associated with the IRAK1 gene expression [59]. Overall, several studies have reported the altered expression of TLRs (TLRs 1/2, TLRs 4-10) in people with obesity and/or T2D [25,26,27].

LncRNAs are emerging as key players in inflammatory and innate immune signaling cascades and are involved in pre- and post-transcriptional gene regulation [60,61]. Many lncRNAs are known to be differentially expressed in different tissues from individuals with obesity. SRA1 expression is more notable in the tissues or organs that have high energy demand such as heart, adipose tissue, skeletal muscle, and liver [36,62,63]. Furthermore, in these tissues/organs, SRA1 regulates several pathophysiological processes such as myocyte/adipocyte differentiation, hepatic steatosis, stem cell function, steroidogenesis, mammary gland development, and tumorigenesis [64]. Increased *SRA1* expression has been reported, in order, in adipocyte fractions from white adipose tissue, brown adipose tissue, and in preadipocytes [40,41]. *SRA1* is involved in regulation of adipocyte differentiation as well as in glucose homeostasis and insulin sensitivity of adipocytes [37,40,41].

Using a mouse model, Liu et al. demonstrated that *SRA1* knockout mice, compared to wild-type controls, had reduced body weight, an increased percentage of lean mass, and less fat percentage, less epididymal/subcutaneous white fat mass, and less liver mass [40]. Two different studies using this mouse model present congruent data showing better insulin sensitivity, less proclivity for obesity development under high-fat diet feeding, reduced hepatic steatosis, and improved glucose tolerance together with less inflammation and lower plasma TNFα levels as well as reduced expression of inflammatory genes (*Tnfα*, and *Ccl2*) in the white fat [40,41]. We also reported that the adipose *SRA1* expression was higher in non-diabetic persons with obesity compared with non-diabetic lean participants, and that the changes associated directly with BMI, PBF, fasting serum insulin, HOMA-IR, certain inflammatory markers but inversely with HbA1c level [42]. The results from previous studies collectively suggest that SRA1 may play a role in the adipose tissue function, pathobiology, as well as in inflammatory cascades implicated with insulin resistance, presenting it as a new target for therapeutic intervention of metabolic inflammation and insulin resistance.

Our data further show significant association of *SRA1* expression with that of TLR3, TLR4, TLR7, TLR9, NF-κB, IRF5, MyD88, and TRAF6 in the adipose tissue. Interestingly, TLR9 and TRAF6 associated differentially with the adipose *SRA1* expression depending on the diabetic status: TLR9/TRAF6 associated positively with *SRA1* expression in individuals with T2D but associated negatively with *SRA1* expression in individuals without T2D. We speculate that the negative association between these factors and adipose *SRA1* expression in the absence of diabetes factor imparts tenacity to the positive relationship of these factors with *SRA1* expression in the cohort with diabetes. These observations need to be further validated in larger cohorts with more diverse populations.

Our data further show that TLR3/TLR9 expression was associated independently with the expression of *SRA1* in individuals with T2D, while TLR3, TLR7 and IRAK1 were identified as the independent predictors of adipose *SRA1* expression in individuals with obesity. Taken together, TLR3 remains the independent predictor of adipose *SRA1* expression in settings of both obesity and T2D. Overall, these data support the plausibility of adipose *SRA1* expression to be considered as a novel, surrogate biomarker of adipose inflammation in obesity/T2D. However, caution will be needed for interpreting results of this primarily correlative study and further investigations will be required to determine: (i) which immunometabolic insults such as adipocytokines, chemokines, as well as gluco-lipotoxic and oxidative stresses may lead to the induction or upregulation of adipose *SRA1* expression, (ii) how does the *SRA1* expression differ between adipocytes and stromal vascular fraction, especially in monocytes/macrophages, and (iii) how will *SRA1* gene silencing and overexpression in adipocytes and monocytes/macrophages affect the insulin-stimulated glucose uptake as a measure of cellular function modulation.

## 5. Conclusions

In conclusion, our data show that the AT *SRA1* expression correlates with TLRs-3,4,7,9, MyD88, NF-κB, and IRF5 expression in individuals with T2D and with TLR2, IRAK1, and IRF3 expression in individuals with obesity. AT-*SRA1* expression was independently predicted by TLR3/TLR7 and IRAK1 in those with obesity and by TLR3/TLR9 in individuals with T2D. An association between the increased *SRA1* expression in the adipose tissue and the markers of immune signaling derangement in this compartment suggest that SRA1 molecules may have significance as new surrogate biomarkers of metabolic inflammation.

## Figures and Tables

**Figure 1 cells-11-04007-f001:**
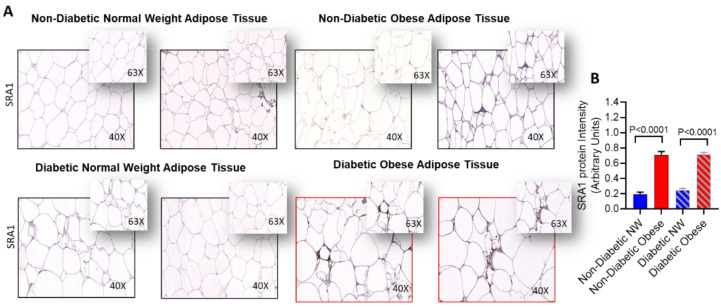
Adipose SRA1 protein expression in adipose tissue. SRA1 protein expression was determined in adipose tissue samples from non-diabetic individuals with NW and obesity, as well as from persons with type-2 diabetic (T2D), as NW and those with obesity, 10 each, using by immunohistochemistry (IHC) as described in Materials and Methods. IHC staining intensity was expressed as arbitrary units (AU) and the data (mean ± SEM) were compared between NW and obese populations, with or without T2D, using unpaired *t*-test and *p* < 0.05 was considered significant. (**A**) The representative IHC images are shown for non-diabetic NW/obese and T2D NW/obese individuals, one each. (**B**) IHC staining data (AU) show the elevated adipose SRA1 expression in persons with obesity compared to NW persons, whether with or without T2D (*p* < 0.0001).

**Figure 2 cells-11-04007-f002:**
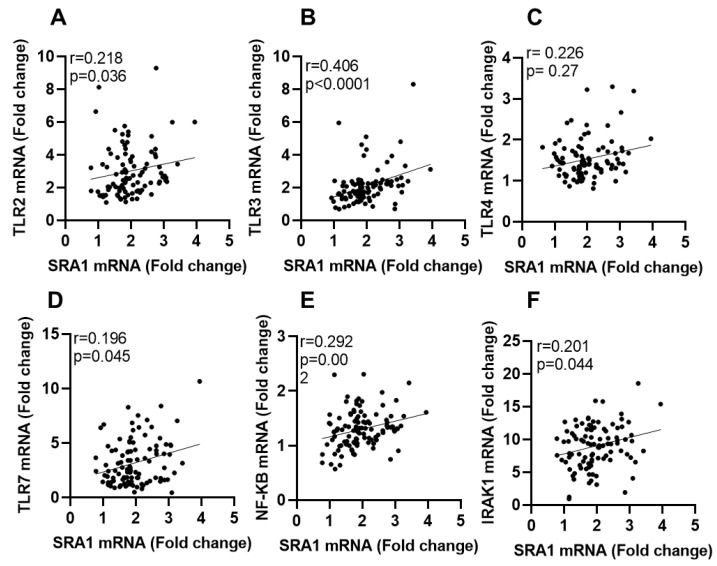
Adipose tissue SRA1 gene expression is correlated with TLRs and signaling molecules. Subcutaneous adipose tissues were obtained from lean, overweight, obese individuals. In this case, mRNA expression of SRA1, TLR2, TLR3, TLR4, TLR7, IRAK1 and NF-kB was detected by real-time RT-PCR and represented as fold change over controls. (**A**–**F**) In the studied population, SRA1 transcript levels, in adipose tissue, are positively correlated with TLRs and their signaling molecules.

**Figure 3 cells-11-04007-f003:**
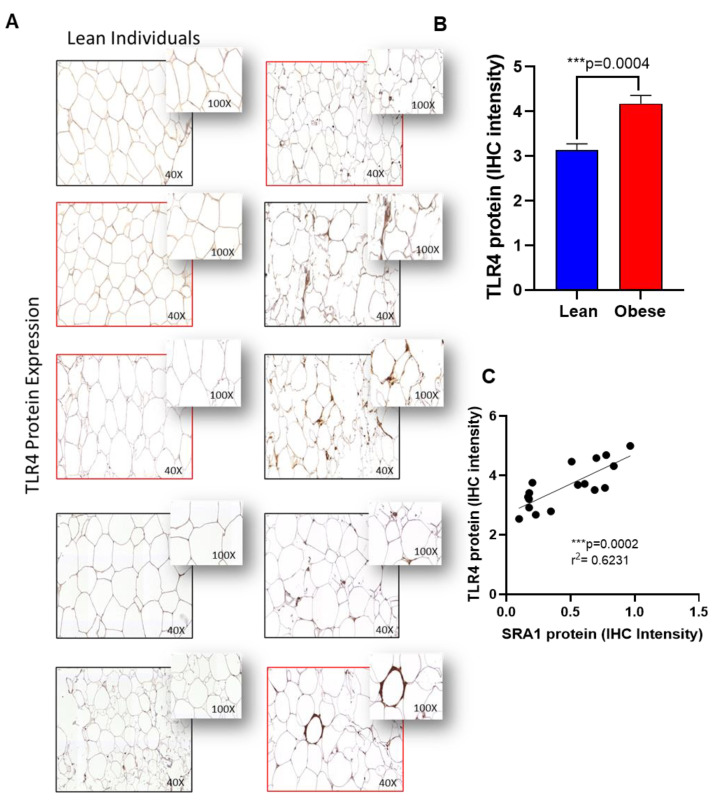
Adipose TLR4 protein expression in adipose tissue and its correlation with SRA1 protein expression. TLR4 protein expression was determined in adipose tissue samples from lean (normal weight; NW) and obese individuals (n = 8 for each group) using by immunohistochemistry (IHC) as described in Materials and Methods. IHC staining intensity was expressed as arbitrary units (AU) and the data (mean ± SEM) were compared between NW and obese populations using unpaired *t*-test and *p* < 0.05 was considered significant. (**A**) The representative IHC images are shown for NW and obese individuals. (**B**) IHC staining intensity data (AU). (**C**) Correlation between TLR4 protein expression with SRA1.

**Figure 4 cells-11-04007-f004:**
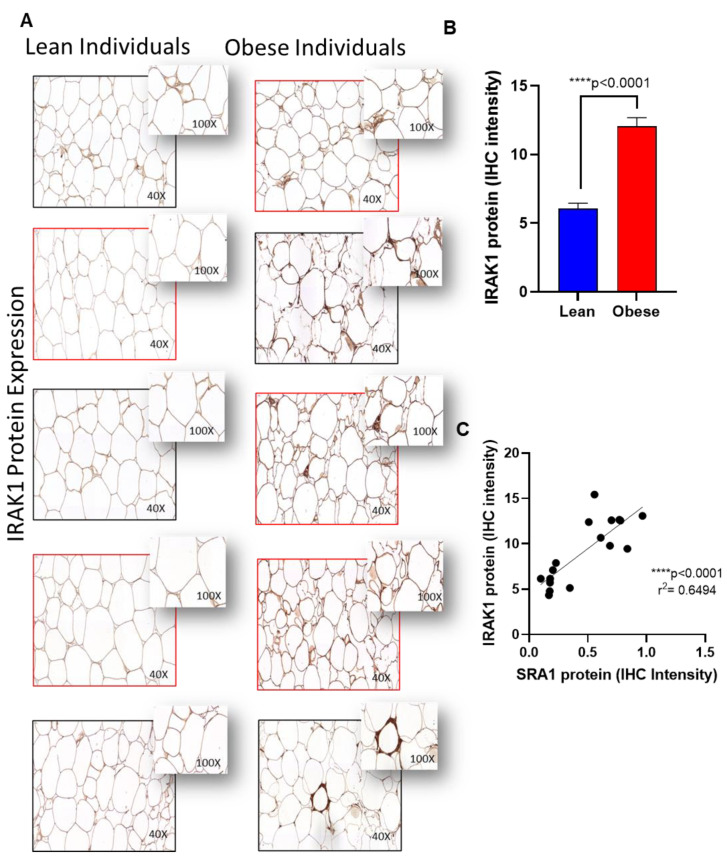
Adipose IRAK1 protein expression in adipose tissue and its correlation with SRA1 protein expression. IRAK1 protein expression was determined in adipose tissue samples from lean (normal weight; NW) and obese individuals (n = 8 for each group) using by immunohistochemistry (IHC) as described in Materials and Methods. IHC staining intensity was expressed as arbitrary units (AU) and the data (mean ± SEM) were compared between NW and obese populations using unpaired *t*-test and *p* < 0.05 was considered significant. (**A**) The representative IHC images are shown for NW and obese individuals. (**B**) IHC staining intensity data (AU). (**C**) Correlation between IRAK1 protein expression with SRA1.

**Figure 5 cells-11-04007-f005:**
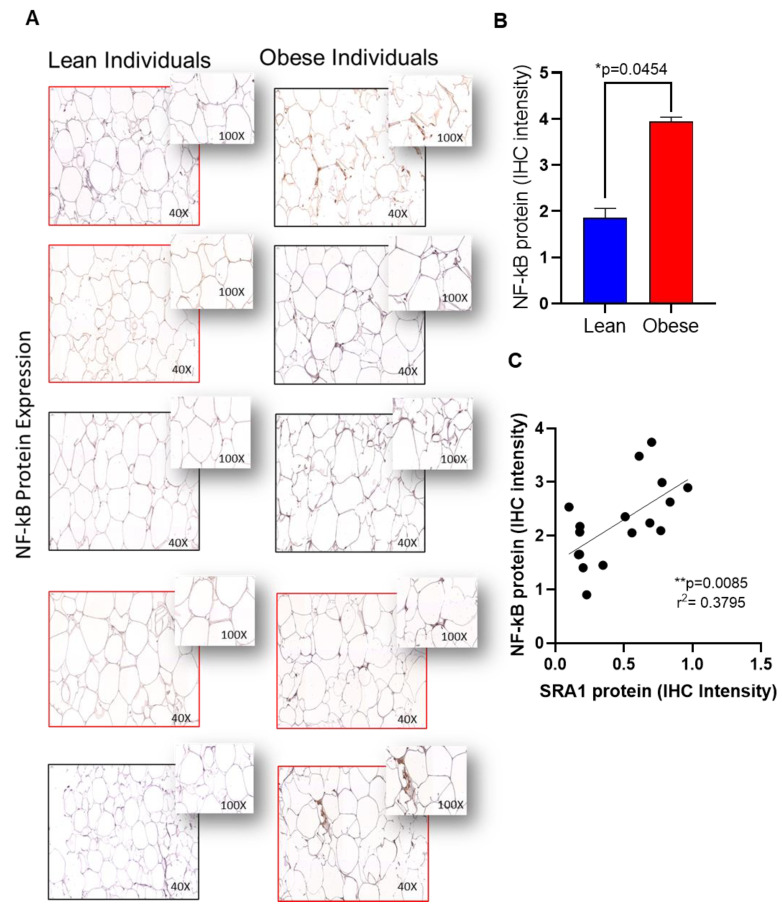
Adipose NF-kB protein expression in adipose tissue and its correlation with SRA1 protein expression. NF-kB protein expression was determined in adipose tissue samples from lean (normal weight; NW) and obese individuals (n= 8 for each group) using by immunohistochemistry (IHC) as described in Materials and Methods. IHC staining intensity was expressed as arbitrary units (AU) and the data (mean ± SEM) were compared between NW and obese populations using unpaired *t*-test and *p* < 0.05 was considered significant. (**A**) The representative IHC images are shown for NW and obese individuals. (**B**) IHC staining intensity data (AU). (**C**) Correlation between NF-kB protein expression with SRA1.

**Figure 6 cells-11-04007-f006:**
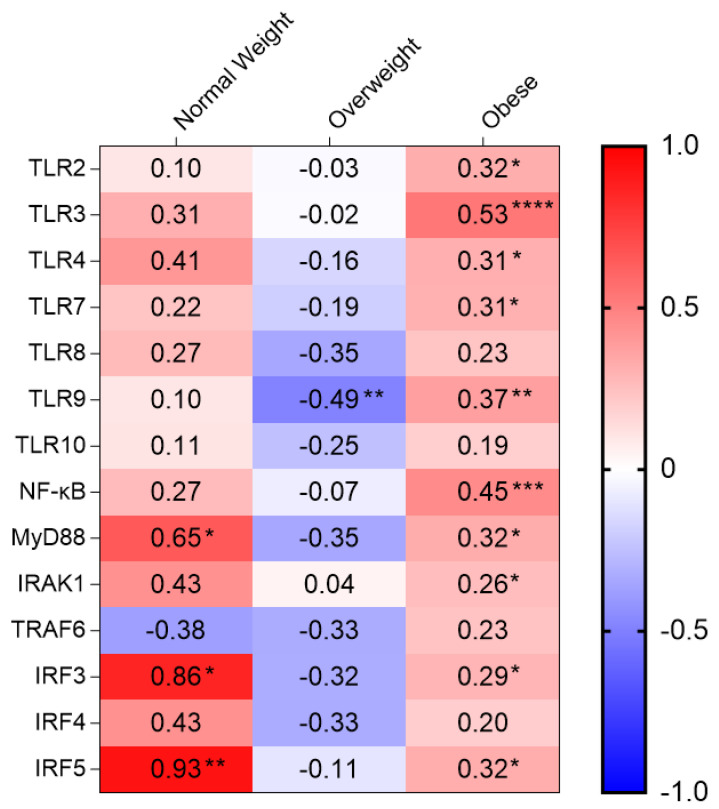
Heat map of the correlation of SRA1 expression with TLRs and their signaling molecules in adipose tissues obtained from NW, Overweight and Obese. *p* ≤ 0.05 *, *p* ˂ 0.01 **, *p* ˂ 0.001 ***, *p* ˂ 0.0001 ****.

**Figure 7 cells-11-04007-f007:**
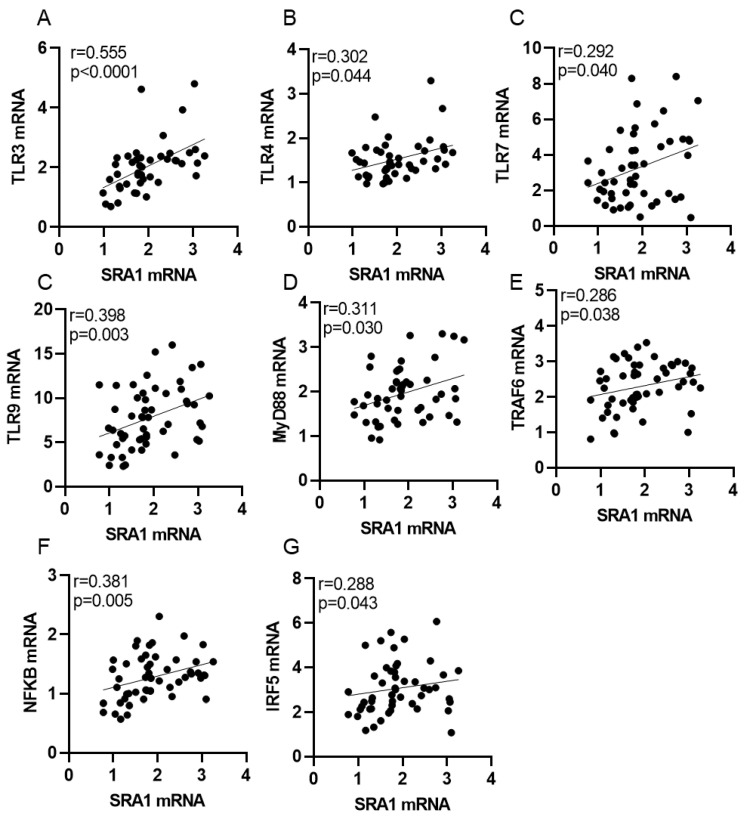
SRA1 gene expression is correlated with TLR3, TLR4, TLR7, TLR9, Myd88, TRAF6, NF-kB or IRF5 in the adipose tissues obtained from individuals with T2D (**A**–**G**).

**Table 1 cells-11-04007-t001:** Adipose expression (mRNA) of TLRs, their downstream signaling mediators, IRFs, and SRA1 in participants with and without T2D.

Inflammatory Marker	Total Participants (N = 108)										
	Without T2D (N = 55)					With T2D (N = 53)					
	Normal Weight (N = 8)	Overweight (N = 19)	Obese (N = 28)	Normal Weight vs. Overweight (*p* Value) #	Normal Weight vs. Obese (*p* Value) #	Normal Weight (N = 4)	Overweight (N = 13)	Obese (N = 36)	Normal Weight vs. Overweight (*p* Value) #	Normal Weight vs. Obese (*p* Value) #	Without T2D (N = 55) vs. With T2D (N = 53) (*p* Value) $
SRA1	1.7 ± 0.24	1.8 ± 0.61	2.2 ± 0.62	0.335	0.025 *	2.3 ± 0.95	1.7 ± 0.55	1.9 ± 1.17	0.175	0.436	0.373
TLR2	1.95 ± 1.04	2.88 ± 1.96	3.20 ± 1.31	0.197	0.018 *	2.18 ± 0.76	2.82 ± 1.17	3.27 ± 1.66	0.465	0.272	0.305
TLR3	1.49 ± 0.30	1.79 ± 1.17	2.57 ± 1.50	0.629	0.019 *	3.24 ± 2.20	1.74 ± 1.06	2.07 ± 0.59	0.168	0.616	0.618
TLR4	1.31 ± 0.23	1.42 ± 0.41	1.67 ± 0.65	0.661	0.267	2.04 ± 0.89	1.33 ± 0.31	1.57 ±0.45	0.144	0.441	0.609
TLR7	2.06 ± 0.83	3.15 ± 2.27	3.38 ± 2.17	0.502	0.141	2.89 ± 1.51	1.92 ± 1.36	3.78 ± 2.04	0.387	0.638	0.546
TLR8	1.68 ± 0.69	3.51 ± 2.90	3.31 ± 1.96	0.117	0.061 *	2.85 ± 0.75	2.71 ± 1.60	3.94 ± 2.78	0.776	0.655	0.202
TLR9	8.51 ± 3.91	7.85 ± 3.37	7.32 ± 3.87	0.803	0.292	7.02 ± 5.62	7.63 ± 2.54	7.91 ± 3.47	0.379	0.357	0.852
TLR10	21.88 ± 22.28	23.04 ± 22.15	29.88 ± 37.15	0.719	0.549	24.96 ± 15.52	23.97 ± 24.62	23.05 ± 20.07	0.625	0.65	0.976
NF-κB	1.18 ± 0.20	1.28 ± 0.33	1.38 ± 0.28	0.476	0.053	1.47 ± 0.29	1.28 ± 0.42	1.26 ± 0.37	0.342	0.282	0.559
MyD88	1.62 ± 0.47	1.86 ± 0.73	1.85 ± 0.57	0.421	0.346	2.71 ± 0.77	1.96 ± 0.57	1.92 ± 0.59	0.166	0.112	0.240
IRAK1	6.14 ± 2.83	8.19 ± 3.37	9.12 ± 3.28	0.119	0.042 *	8.01 ± 1.11	9.45 ± 3.46	9.69 ± 3.23	0.432	0.353	0.116
TRAF6	2.49 ± 0.35	2.26 ± 1.18	2.36 ± 0.98	0.113	0.255	2.12 ± 0.80	2.26 ± 0.75	2.33 ± 0.64	0.922	0.712	0.544
IRF3	1.81 ± 0.57	2.31 ± 0.68	2.54 ± 0.61	0.152	0.065	2.45 ± 0.80	2.73 ± 0.56	2.65 ± 0.59	0.4542	0.675	0.047 *
IRF4	2.21 ± 0.84	3.61 ± 2.18	3.40 ± 1.99	0.228	0.169	3.74 ± 3.41	4.32 ± 2.10	3.95 ±1.46	0.702	0.828	0.020 *
IRF5	1.53 ± 0.33	2.82 ± 1.44	2.73 ± 1.08	0.027 *	0.016 *	2.72 ± 0.93	2.90 ± 1.31	3.13 ±1.13	0.879	0.584	0.039 *

# = One way Anova, Kruskal-Wallis test, $ = Mann-Whitney test. * Significant.

**Table 2 cells-11-04007-t002:** Spearman’s correlation analysis of adipose gene expression of *SRA1* with TLRs, their signaling mediators, and IRFs in the total study population.

All participants (N = 108)		
Adipose Marker	*r*	*p* Value
TLR2	0.218	0.036 *
TLR3	0.406	<0.0001 ****
TLR4	0.226	0.027 *
TLR7	0.196	0.045 *
TLR8	0.071	0.481
TLR9	0.080	0.411
TLR10	0.074	0.459
NF-κB	0.297	0.002 **
MyD88	0.149	0.134
IRAK1	0.201	0.044 *
TRAF6	0.032	0.746
IRF3	0.139	0.187
IRF4	−0.009	0.931
IRF5	0.175	0.080

*p* ≤ 0.05 *, *p* ˂ 0.01 **, *p* ˂ 0.0001 ****.

**Table 3 cells-11-04007-t003:** Spearman’s correlation analysis of adipose gene expression of *SRA1* with TLRs, their signaling mediators, and IRFs in individuals differing by obesity levels.

Obesity Level	Normal Weight		Overweight		Obese	
	(N = 12)		(N = 32)		(N = 64)	
Adipose Marker	*r*	*p* Value	*r*	*p* Value	*r*	*p* Value
TLR2	0.095	0.823	−0.026	0.895	0.317	0.017 *
TLR3	0.310	0.456	−0.016	0.934	0.531	<0.0001 ****
TLR4	0.405	0.320	−0.164	0.386	0.311	0.022 *
TLR7	0.224	0.533	−0.190	0.297	0.305	0.015 *
TLR8	0.267	0.488	−0.347	0.060	0.229	0.071
TLR9	0.098	0.762	−0.489	0.005 **	0.374	0.002 **
TLR10	0.105	0.746	−0.248	0.194	0.192	0.134
NF-κB	0.266	0.404	−0.071	0.699	0.454	<0.001 ***
MyD88	0.648	0.043 *	−0.347	0.060	0.324	0.010 *
IRAK1	0.429	0.289	0.044	0.818	0.255	0.044 *
TRAF6	−0.378	0.226	−0.327	0.073	0.233	0.064
IRF3	0.857	0.014 *	−0.322	0.088	0.290	0.030 *
IRF4	0.429	0.397	−0.330	0.075	0.195	0.136
IRF5	0.929	0.003 **	−0.105	0.575	0.321	0.010 *

*p* ≤ 0.05 *, *p* ˂ 0.01 **, *p* ˂ 0.001 ***, *p* ˂ 0.0001 ****.

**Table 4 cells-11-04007-t004:** Spearman’s correlation analysis of adipose gene expression of *SRA1* with TLRs, their signaling mediators, and IRFs in individuals with/without T2D.

Diabetes Status	Without T2D		With T2D	
	(N = 55)		(N = 53)	
Adipose Marker	*r*	*p* Value	*r*	*p* Value
TLR2	0.206	0.165	0.223	0.136
TLR3	0.247	0.080	0.555	<0.0001 ****
TLR4	0.096	0.522	0.302	0.044 *
TLR7	0.140	0.309	0.292	0.040 *
TLR8	0.028	0.843	0.180	0.210
TLR9	−0.290	0.034 *	0.398	0.003 **
TLR10	0.062	0.667	0.084	0.556
NF-κB	0.191	0.162	0.381	0.005 **
MyD88	0.014	0.918	0.311	0.030 *
IRAK1	0.220	0.117	0.239	0.099
TRAF6	−0.318	0.019 *	0.286	0.038 *
IRF3	0.051	0.739	0.220	0.137
IRF4	0.066	0.649	−0.017	0.909
IRF5	0.167	0.243	0.288	0.043 *

*p* ≤ 0.05 *, *p* ˂ 0.01 **, *p* ˂ 0.0001 ****.

**Table 5 cells-11-04007-t005:** The multiple linear regression analysis.

All participants (N = 108)	ANOVA	r^2^ = 0.21		*p* value < 0.0001
	Predictor Variable	TLR3	β value = 0.392	*p* value < 0.0001
		IRAK1	β value = 0.279	*p* value = 0004
With T2D (N = 53)	ANOVA	r^2^= 0.35		*p* value < 0.0001
	Predictor Variable	TLR3	β value = 0.499	*p* value < 0.001
		TLR9	β value = 0.329	*p* value = 0.010
Normal Weight (N = 12)	ANOVA	r^2^= 0.64		*p* value = 0.005
	Predictor Variable	MyD88	β value = 0.801	*p* value = 0.005
Overweight (N = 32)	ANOVA	r^2^= 0.25		*p* value = 0.004
	Predictor Variable	TLR9	β value = −0.499	*p* value = 0.004
Obese (N = 64)	ANOVA	r^2^= 0.38		*p* value < 0.0001
	Predictor Variable	TLR3	β value = 0.464	*p* value < 0.0001
		IRAK1	β value = 0.317	*p* value = 0.005
		TLR7	β value = 0.246	*p* value = 0.027

## Data Availability

The data presented in this study are available on request from the corresponding author.

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
