# Peer review of "Expression of Steroid Receptor RNA Activator 1 (SRA1) in the Adipose Tissue Is Associated with TLRs and IRFs in Diabesity"

_cells, 2022, doi:10.3390/cells11244007_

Round 1
Reviewer 1 Report (Previous Reviewer 2)
The manuscript has been improved. Only some minors should be changed, sentence with two periods, p or P, r or R values.
Author Response
Response to Reviewer 1 Comments
Ref: Revised Submission Manuscript ID: cells-2046634
We thank the reviewer for encouraging comments and approving the manuscript.
comments and Suggestions for Authors
The manuscript has been improved. Only some minors should be changed, sentence with two periods, p or P, r or R values.
Authors’ response: All “p” and “r” values have been corrected in the text and Figs for uniformity.

Reviewer 2 Report (Previous Reviewer 3)
Now that the authors have revised the manuscript and have responded to my comments, the manuscript looks good.
Author Response
Response to Reviewer 2 Comments
Ref: Revised Submission Manuscript ID: cells-2046634
We thank the reviewer for accepting and approving our revised manuscript.
Comments and Suggestions for Authors
Now that the authors have revised the manuscript and have responded to my comments, the manuscript looks good.
Thank you for accepting and approving our revised manuscript.

Reviewer 3 Report (New Reviewer)
The authors of this study performed systemic analysis to identify the relationship between SRA1 and other inflammatory signalings in diabesity patients. This result might be an informative source for the field and might potentially identify therapeutic targets. However, I would not recommend publishing this review article in its current version due to the following major/minor issues.
My major concern goes to the quality of the immunohistochemistry analysis performed in this study. I couldn’t tell the staining specificity in most of the representative images shown in this manuscript. This may also cause by some technical difficulties from adipose tissue sectioning. There are multiple images from this manuscript obtained from defective sections(Figure 1, Diabetic-Obese sample, Figure 3, 4, 5, both samples. The authors need to provide better histological analysis to support their conclusion in this study. Authors may also consider providing H&E stain results as well.
For qPCR analysis in this study, authors need to assess the expression of other mRNAs involved in adipogenesis regulation. For example, authors may check the expression of PPARg, C/EBPa, Plin, Adiponectin in those NW, OW, and obese samples. Correlation analysis between those adipogenic genes, SRA1, TLR should better support the author’s interpretation in this study.
Author Response
Response to Reviewer 3 Comments
We thank the reviewer for your thoughtful comments. Please see below point-by-point response to your comments.
Comments and Suggestions for Authors
The authors of this study performed systemic analysis to identify the relationship between SRA1 and other inflammatory signalings in diabesity patients. This result might be an informative source for the field and might potentially identify therapeutic targets. However, I would not recommend publishing this review article in its current version due to the following major/minor issues.
My major concern goes to the quality of the immunohistochemistry analysis performed in this study. I couldn’t tell the staining specificity in most of the representative images shown in this manuscript. This may also cause by some technical difficulties from adipose tissue sectioning. There are multiple images from this manuscript obtained from defective sections(Figure 1, Diabetic-Obese sample, Figure 3, 4, 5, both samples. The authors need to provide better histological analysis to support their conclusion in this study. Authors may also consider providing H&E stain results as well.
Authors’ response: Thanks for the comments. We have replaced diabetic samples in Figure 1 as suggested (Replaced samples are indicated by red boundary lines)
Figure 1
Figure 3
We have replaced the samples in Figure 3 as suggested (Replaced samples are indicated by red boundary lines)
Figure 4
We have replaced the samples in Figure 4 as suggested (Replaced samples are indicated by red boundary lines)
Figure 5
We have replaced the samples in Figure 5 as suggested (Replaced samples are indicated by red boundary lines)
We have also provided the H&E staining results, as advised (Supplementary files).
For qPCR analysis in this study, authors need to assess the expression of other mRNAs involved in adipogenesis regulation. For example, authors may check the expression of PPARg, C/EBPa, Plin, Adiponectin in those NW, OW, and obese samples. Correlation analysis between those adipogenic genes, SRA1, TLR should better support the author’s interpretation in this study.
Authors’ Response: Thank you for the comments. In this study, our current focus was on the relationship between adipose expression of SRA1 and TLRs and their downstream signaling molecules. Therefore, considering expression of adipogenesis markers and investigate their relationship with SRA1 is topic for a new full-length paper which we will consider in our future work.

Round 2
Reviewer 3 Report (New Reviewer)
None
This manuscript is a resubmission of an earlier submission. The following is a list of the peer review reports and author responses from that submission.
Round 1
Reviewer 1 Report
This manuscript contains ex vivo data describing how Steroid Receptor RNA Activator 1 expression correlates with the mRNA expression of inflammation related genes in control individuals and patients with obesity with or without type 2 diabetes mellitus. The limitations of the study is that it lacks functional experiments. Otherwise, the manuscript presents clinically important data and is well written.
Specific suggestions:
1. Due to the current preference in scientific literature of a non-stigmatizing language to describe diseases, adjectives should be avoided. Therefore, e.g. instead of "obese patient", "patient with obesity" should be written. Please, amend this in lanes 28, 29, 92, 100, 203, 208, etc.
2. IL-6 is a pleiotropic cytokine which has a profound effect on adipocyte browning even in humans. It cannot be considered just as a pro-inflammatory cytokine in this respect.
3. L67: “IL-6” instead of “interleukin (IL)-6”
4. L82: “and T2D” instead of: “andT2D”
5. HbA1c should be defined in lane 112.
6. L158: “dilution” instead of “diluion”
7. L194: The unpaired t-test was not mentioned in 2.6. Did the Authors check normality distribution of the data?
8. The legend of Table 5 should be more detained.
9. In Table 5, the data of 4th-5th row is missing.
10. Page 9. “LncRNAs” instead of “Long noncoding RNAs (lncRNAs)”. This sentence also requires reference(s).
Reviewer 2 Report
In this study, the authors investigated the association between the expression of steroid receptor RNA activator 1 (SRA1) with TLRs and IRFs in adipose tissue. SRA1 has been reported to be increased in obesity, however, the authors claimed to be the first time to report it. TLRs and IRFs are associated with inflammation, and SRA1 will be associated with the expression of TLRs and IRFs. The main concern, this study lacks protein expression of TLRs and IRFs without potential mechanisms.
Other minors: TNF abbreviations show twice, and some abbreviations were not given full names.
Number and unit are not separated, such as 0.5g.
Supplementary table 1 was not given.
Reviewer 3 Report
In this study, the authors describe the association between steroid receptor RNA activator 1 (SRA1) expression and TLRs and IRFs in adipose tissue in healthy control individuals and patients with obesity with or without type 2 diabetes mellitus. The authors present well-written and clinically important data. However, functional experiments are suggested to be included.
- Supplementary table 1 was not given.
- The TNF abbreviations were shown twice “tumor necrosis factor (TNF)” row 61 and 67. The same for interleukin rows 60 and 67. No full name was given for the abbreviation “NF-κB”, “CXCL”, “TGFβ”, NF-κB
- In row 82 “andT2D” should be separated. Number and unit are not separated throughout the manuscript.
- There should be consistency in the manuscript. In some rows, it has been written 75 % with space between the number and the %-sign and in some rows, there is no space.
- HbA1c levels of the individuals should be defined in the Material and Methods section.
- Instead of writing “obese people”, it should be written “individuals with obesity” or “patients with obesity”.
- It would be nice to see some of the results in figures instead of tables. It is easier to interpret the results in a figure.
- References are needed in the Discussion on page 9 “Long noncoding RNAs (lncRNAs) are emerging as key plyers in inflammatory signaling cascades and are involved in pre- and post-transcriptional gene regulation”.
- "players" instead of "plyers" in the #8.
- The authors write “In the present study, we demonstrate for the first time to the best of our knowledge that the adipose tissue SRA1 protein expression is significantly increased in individuals with obesity, as compared to their normal weight counterparts, regardless of T2D status”. However, they have previously shown this in their published article “Adipose Tissue Steroid Receptor RNA Activator 1 (SRA1) Expression Is Associated with Obesity, Insulin Resistance, and Inflammation” Cells. 2021 Sep 30;10(10):2602. doi: 10.3390/cells10102602.
- R-value and p-value are missing in Table 5. “Without T2D (N=55)”